# Development of Charge-Augmented Three-Point Water Model (CAIPi3P) for Accurate Simulations of Intrinsically Disordered Proteins

**DOI:** 10.3390/ijms21176166

**Published:** 2020-08-26

**Authors:** Joao V. de Souza, Francesc Sabanés Zariquiey, Agnieszka K. Bronowska

**Affiliations:** 1Chemistry—School of Natural and Environmental Sciences, Newcastle University, Newcastle-upon-Tyne NE1 7RU, UK; j.v.de-souza-cunha2@newcastle.ac.uk (J.V.d.S.); F.Sabanes2@newcastle.ac.uk (F.S.Z.); 2Newcastle University Centre for Cancer, Newcastle University, Newcastle NE1 7RU, UK

**Keywords:** IDPs, water models, molecular dynamics, intrinsically disordered proteins, CAIPi3P

## Abstract

Intrinsically disordered proteins (IDPs) are molecules without a fixed tertiary structure, exerting crucial roles in cellular signalling, growth and molecular recognition events. Due to their high plasticity, IDPs are very challenging in experimental and computational structural studies. To provide detailed atomic insight in IDPs’ dynamics governing their functional mechanisms, all-atom molecular dynamics (MD) simulations are widely employed. However, the current generalist force fields and solvent models are unable to generate satisfactory ensembles for IDPs when compared to existing experimental data. In this work, we present a new solvation model, denoted as the Charge-Augmented Three-Point Water Model for Intrinsically Disordered Proteins (CAIPi3P). CAIPi3P has been generated by performing a systematic scan of atomic partial charges assigned to the widely popular molecular scaffold of the three-point TIP3P water model. We found that explicit solvent MD simulations employing CAIPi3P solvation considerably improved the small-angle X-ray scattering (SAXS) scattering profiles for three different IDPs. Not surprisingly, this improvement was further enhanced by using CAIPi3P water in combination with the protein force field parametrized for IDPs. We also demonstrated the applicability of CAIPi3P to molecular systems containing structured as well as intrinsically disordered regions/domains. Our results highlight the crucial importance of solvent effects for generating molecular ensembles of IDPs which reproduce the experimental data available. Hence, we conclude that our newly developed CAIPi3P solvation model is a valuable tool for molecular simulations of intrinsically disordered proteins and assessing their molecular dynamics.

## 1. Introduction

Atomistic molecular dynamics (MD) simulations can reliably assess dynamical properties in equilibrium structures of molecular systems of interest, given an ergodic sampling and an accurate force field. The force field parameters are calibrated to reproduce properties measured by experiments or simulations. Considering the immense complexity of macromolecular systems, and the sensitivity of weak (hydrogen-bonding and dispersion) non-covalent interactions in a liquid phase, contributing to intra-solute, solute-solvent and solvent-solvent interactions, even modest inaccuracies in models and their parameters can adversely impact the results of atomistic molecular simulations, especially of challenging systems such as intrinsically disordered proteins (IDPs). IDPs are elusive to experimental studies, thus atomistic simulations are a crucial tool to provide detailed insight into their complex structure, dynamics, and function. Unfortunately, computational studies of IDPs are often found to disagree with experimental data. The free energy landscape of IDPs is inverted compared to the structured proteins [1], which makes computational studies focusing on IDPs very challenging. Discrepancies between theory and experiments are commonly attributed to either force field biases [2,3] or insufficient sampling. This motivated the development of molecular force fields designed to handle IDPs [4,5,6] and to apply enhanced sampling techniques [7,8,9,10,11] or restraints derived from experimental data (e.g., solution NMR) in simulations of IDPs [10,12,13]. The outcomes of those efforts were successful to various extents; however, IDP simulations still require parameter improvements [10,14,15,16,17]

IDPs have unordered structures in aqueous solution, and while either dehydrated or interacting with lipid membranes, they exhibit increased amounts of ordered secondary structures [18]. This clearly shows that IDPs are highly sensitive to solvation effects [19,20] and suggests that focusing on the improvement of the water models used in the simulations may offer a more accurate yet computationally feasible framework for reliable simulations of this class of proteins.

The complexity of the water properties, combined with multiple possible levels of approximation, has led to the proposal of dozens of water models. Simplified classical water models, such as widely popular three-point SPC [21] and TIP3P [22] models, are currently indispensable components of atomistic MD toolkits. Yet, despite several decades of extensive research, these models are still far from perfect. To start, none of them accurately reproduces the key properties of bulk water [23]. Alternative approaches, most notably the “optimal” three-charge, four-point rigid water model (OPC) [24] have been developed and tested recently. OPC uses the optimised distribution of point charges to best describe the electrostatics of the water molecule, in contrast to the ‘conventional’ approach to constructing the classical solvation models, which often imposes geometry constraints [25]. However, simplified classical water models, particularly the simplest, non-polarisable three-point models, are still the most commonly used in the biomolecular simulations community, due to their computational efficiency and simplicity.

In simulations of IDPs, the best-performing water models have a charge distribution with a large dipole moment, a large quadrupole moment, and negative charge out of the molecular plane, to give symmetrically ordered tetrahedral hydration [26]. We have observed that the dipole calculated for the very popular TIP3P model is too low, resembling a dipole of an isolated water molecule in a vacuum (2.36D), rather than of a dipole in the liquid bulk state (3D). The exact value of the liquid water dipole is still debated; however, in this study, we rely on the results of most recent first-principles simulations of liquid state water. Nevertheless, to improve the properties of the TIP3P water model, it seemed crucial to adjust the dipole: we have done so by augmenting partial atomic charges of the water molecule. The performance of such an improved model, denoted as the Charge-Augmented Three-Point Water Model for Intrinsically Disordered Proteins (CAIPi3P), was subsequently tested on model IDPs: histatin 5, R/S-peptide, partially disordered *At2g23090* protein from *A. thaliana*, and two domains of the La-related protein: RNA recognition Motif 1 (RRM1) and La-Motif (LaM). We observed that the dipole adjustment dramatically improved the performance of the model, in terms of the reproducibility of experimental data for IDPs, without negatively affecting the performance, speed, and data reproducibility for the folded regions/domains of partially disordered systems or the performance and data reproducibility for the folded regions/domains of partially disordered systems or the ‘structured’ proteins.

## 2. Results

### 2.1. Parametrisation of CAIPI3P Water Model

Unlike globular proteins, intrinsically disordered proteins (IDPs) do not have a proper hydrophobic core. As such, long-range electrostatic interactions play an important role in defining of IDP behaviour [19,20]. Therefore, to accurately predict the dynamics of IDPs from the atomistic molecular simulations, the water interactions with the environment and with the solute of interest need to be re-calibrated.

To develop an improved solvent model for simulations of IDPs, a systematic scan of dipole moment and partial atomic charges assigned on the molecular scaffold of the popular TIP3P framework [22]. Different dipole moments were tested using histatin 5 as a reference system. The model that showed the best agreement with experimental small-angle X-ray scattering (SAXS) pair distance distribution function (PDDF) was selected for CAIPi3P. The atomic charges and the value for the dipole moment are shown in Table 1. The CAIPi3P model was developed to scale electrostatic interactions between all molecular components interacting with the solvent. Nonetheless, CAIPi3P modifies the charge values, primarily affecting only the coulombic interactions, for both solvent–solvent and solvent–solute interactions. This approach is based on a similar idea to that of solute–solvent interactions through Lennard-Jones parameters, which was the basis for the creation of the AMBER03ws force field, which was developed for IDPs simulations and was designed to be fully compatible with the four-point TIP4P/2005 water model [27,28]. The optimisation curve for different values of dipole moment is shown in Appendix A.

### 2.2. MD Simulations of a Full-Length IDP: Histatin 5

Histatin 5 belongs to a family of well-characterised antimicrobial peptides secreted in human salivary (submandibular) glands [2,30]. It is a highly water-soluble IDP that has been used as a model in many computational studies [3,30]. Although there is no experimental structure of histatin 5 available to date, two research groups: Henriques and coworkers [30] and Cragnell and coworkers [31] published its SAXS data. In their studies, the best agreement between simulations and experimental results was achieved using the AMBER03ws force field with the TIP4P/2005 water model. 

Our simulations employing AMBER03ws resulted in more accurate sampling than when using AMBER99SB-ILDN when compared to the experimental SAXS scattering profile (Figure 1). This improved even further when combined with the CAIPi3P solvation model (Figure 1). The combination of AMBER03ws (protein) and CAIPi3P (solvent) outperformed AMBER03ws+TIP4P/2005: the root-mean-square difference (RMSD) between experimental and calculated PDDF RMSD_exp-calc_ was = 0.01, with a χ^2^ for the I(q) of 1.1 for AMBER03ws+TIP4P/2005, in comparison with RMSD_exp-calc_ = 0.007 and a χ^2^ for the I(q) of 0.4 for the AMBER03ws+CAIPi3P. It is important to remark that AMBER03ws+TIP4P/2005 is considered a highly accurate combination of water model and force field for simulating histatin 5 [30]. 

Simulations using the AMBER99SB-ILDN+OPC combination resulted in a scattering profile comparable to AMBER99SB-ILDN+TIP3P (Figure 1, orange and blue curves respectively). With χ^2^ for the I(q) of 3.1 for AMBER99SB-ILDN+OPC, the most accessed conformation was a collapsed conformation with a radius of gyration of 0.9 nm, underperforming in comparison to the AMBER99SB-ILDN+CAIPI3P combination. Even though the AMBER03ws+CAIPi3P combination yielded the best agreement with the experimental data, the improvement resulting from the application of CAIPi3P was apparent, regardless of the protein force field used (Figure 1; red and purple curves). These results are very encouraging in the context of the transferability of the CAIPi3P model and its applicability to studies of intrinsically disordered macromolecules.

As shown in Table 2, both the AMBER03ws and AMBER99SB-ILDN force fields attained reasonable sampling of the experimental radii of gyration. The AMBER03ws combined with CAIPi3P had its distribution peak around 1.4 nm, sampling more expanded conformations than any of the combined sets.

The solute–solvent and solvent–solvent long-range electrostatic interactions play a significant role in defining of the conformational landscape of IDPs. The solvation model is, therefore, crucial for the sufficient sampling of the IDPs. Figure 2 shows that two clusters obtained by the simulations using CAIPi3P, calculated from the RMSD matrix, are a very similar one to another. TIP4P/2005, on the other hand, sampled two sparse conformations, with the system collapsed on itself for nearly half of the simulation time. The most compact (self-collapsed) conformation affected the PDDF, which resulted in the ensemble with the radial distribution resembling that of a globular protein, which directly affects the calculated sample, resulting in the Gaussian-like distribution for the SAXS PDDF. (Figure 1).

The internal potential energy for the histatin 5 increased when using CAIPi3P (Figure 3), showing that protein intramolecular interactions should be disrupted, and the solute–solvent interactions should be increased (Figure 3). Although AMBER03ws can increase the radius of gyration by increasing the solute–solvent contribution, CAIPi3P simulations resulted in correct configurations by increasing the Coulombic solute–solvent and solvent–solvent interactions, increasing the structural potential energy, as can be seen in the red cluster in Figure 3, which has both a high value of structural energy and radius of gyration.

### 2.3. The CAIPi3P Effect on the Sampling of the Charged Repeats of R/S-Peptide

Arginine–serine repeats (R/S repeats) play an essential role in cellular signalling since the phosphorylation of the serine residues is crucial for the regulation of many enzymes and receptors. Because of the accumulation of highly polar arginine and serine residues, intrinsically disordered R/S-peptide is highly polar itself. As such, it presents a challenging IDP to be correctly modelled. Several studies on its dynamics have been performed, using solution NMR and SAXS [32,33].

The calculated SAXS parameters and radii of gyration for the R/S peptide and their comparison to the experimental data available are shown in Figure 4. The choice of the protein force field played a critical role in reproducing the experimental data (Figure 4). Application of both TIP4P and CAIPi3P water models reproduced the experimental SAXS PDDF and scattering profile (Figure 4), but simulations performed using the CAIPi3P model achieved an average radius of gyration in better agreement with the experimental values (Table 2).

Regardless of the solvation model used, the R/S peptide simulated with the AMBER99SB-ILDN force field collapsed on itself after 30 ns of simulation, resulting in a very different distribution when compared to the experimental data. (CAIPi3P PDDF RMSD_exp-calc_ = 0.033, I(q) χ^2^ = 5.6; and TIP3P RMSD_exp-calc_ = 0.034, I(q) χ^2^ = 5.7 respectively; Figure 4). AMBER99SB-ILDN sampled conformations with a better agreement with the experimental curve states when used combined with the OPC water model (I(q) χ^2^ = 3.3). Employing AMBER03ws force field improved the agreement with the experimental data, regardless of the water model (TIP4P/2005 RSMDexp-calc = 0.005, I(q) χ^2^ = 1.4 and CAIPi3P RMSDexp-calc = 0.003, I(q) χ^2^ = 1.3). Nevertheless, the production trajectories obtained with CAIPi3P water showed an average radius of gyration within the experimental range of the radius of gyration (Table 2), resulting in a predicted average radius of gyration of 1.3 nm.

For the R/S peptide sampling assessment, two clusters of each combination of protein force field/water molecule were selected to visual inspection. CAIPi3P clusters remained in an opened, extended conformation for approximately 85% of the simulation run (white area around the red square, Figure 5A). Both solvent models enabled interactions between the N-terminal region and the 16 residues R/S repeat region. In the TIP4P/2005 model, a partial collapse occurred early in the simulation, and it is highlighted by the TIP4P/2005 red ensemble in Figure 5 (lower panel).

Since the R/S repeat region is highly polar (Figure 5; highlighted regions in yellow), it might be expected for this region to interact with water favourably. The glycine residue, which is adjacent to the R/S repeat, acts as a “hinge”, partially collapsing the ensemble (blue clusters in Figure 5) in simulations using both solvent models. Considering this structural peculiarity, R/S peptide presents itself a challenge for modelling and suffers more from the force-field selection from the solvation model, since the force-fields are known to be directly affected by the accuracy of the calculated charges. The results show that there are improvements still to be made on the AMBER force field and CAIPi3P parameters, yet the sampling achieved by the application of CAIPi3P model outperformed that of TIP4P/2005, as shown with the sampled SAXS curves. 

### 2.4. The Effects of CAIPi3P on Partially Disordered Structures

The solution NMR structure of the partially disordered protein *At2g23090* from *Arabidopsis thaliana* has been deposited in the RCSB PDB Data Bank (PDB code: 1VWK [34]). It was used to assess the accuracy of the CAIPi3P water model for very flexible and partially disordered proteins since it has a C-terminal globular region and a long loop formed by 46 residues. While *At2g23090* presents itself as a challenging benchmarking test, the NMR ensemble was used to study the possible dynamics. The AMBER99SB-ILDN protein force field combined either with CAIPi3P or OPC water models outperformed all other combinations of protein force fields and solvent models (Figure 6), with an I(q) RMSDexp-calc = 0.08 and an I(q) χ^2^ = 1.1 for AMBER99SB-ILDN+CAIPi3P, and I(q) RMSDexp-calc = 0.02 and an I(q) χ^2^ = 0.4 for AMBER99SB-ILDN+OPC, respectively. This shows that both the CAIPi3P and OPC models obtain accurate ensembles for partially disordered macromolecules (Figure 6, and Table 2 and Table 3) and both models are suitable for simulations for IDPs. This also confirms that the OPC model can accurately sample disordered regions, as shown previously by Shabane and coworkers [24]. 

The AMBER03ws+CAIPi3P combination (Table 2) showed an average of 1.9 nm for its radius of gyration. The protein collapsed on itself, resulting in a Gaussian PDDF with experimental RMSDexp-calc = 0.014, with I(q) χ^2^ = 3.8 for the scattering profile. In contrast, application of the TIP4P/2005 model with the AMBER03ws force field resulted in the unfolding of the globular C-terminal domain (Figure 7), with a radius of gyration centred around 2.6 nm, which is higher than the experimental range. This demonstrates the limitations of applicability of the AMBER03ws force field in the simulations of multi-domain proteins containing globular domains connected by intrinsically disordered regions (IDRs). The C-terminal domain remained folded in simulations using AMBER99SBN-ILDN (Figure 7). The opposite happened for AMBER03ws+TIP4P/2005: the radii of gyration were outside of the experimental range, resulting in average conformations that were too stretched in comparison to the experimental data.

The unfolding of the globular C-terminal domain causes an increase in the structural potential energy in all AMBER03ws simulations (Figure 8). This indicates that AMBER03ws may require improvements for more accurate simulations of proteins containing globular domains. The AMBER99SB-ILDN+CAIPi3P simulation shows sampled values for Rg within the experimental range, with higher potential energy than AMBER99SB-ILDN+TIP3P, which indicates that the protein +CAIPi3P solvent–solute interactions were more favourable, avoiding the self-collapse.

### 2.5. LaRP6-LaM and LaRP6-RRM1

La-related proteins (LaRPs) form a large family of RNA-binding eukaryotic proteins, involved in cell growth and proliferation primarily through the regulation of protein synthesis [35]. All LaRPs are comprised of seven distinct protein families. Other than LaRP6, investigated in this chapter, are LaRP1, LaRP1B, LaRP3 (aka genuine La or SSB), LaRP4A, LaRP4B, and LaRP7 [36,37]. All LaRPs contain the La module, which is a conserved domain for RNA binding. The La module is assembled by two domains: the RNA recognition motif 1 (RRM1) and the La motif (LaM). Their synergistic work regulates the interaction with RNA and the dimer-nucleotide configuration.

Martino and collaborators resolved the structures of both domains of LaRP6 separately using solution NMR. The percentage of residues with a well-defined secondary structure within LaRP6-LaM is 34% and 52% for LaRP6-RRM1 [38]. 

LaRP6-RRM1 has two intra-domain short unstructured loops: one located between A199 and G206, and another one between residues Y250 and E257. LaRP6-RRM1 was less prone to unfold, and it maintained a stable radius of gyration in simulations using AMBER03ws combined with CAIPi3P water model, as shown in Figure 9. AMBER03ws with TIP4P/2005 showed an expanded radius of gyration in comparison to the other three combinations, mainly sampling conformations outside the experimental range (dashed lines in Figure 9). The simulations using CAIPi3P resulted in conformation sampled within the experimental range, and a scattering profile χ^2^ = 0.15 and χ^2^ = 0.2 for AMBER99SB-ILDN+CAIPi3P and AMBER03ws+CAIPi3P, respectively, when compared to the experimental curve (Table 4). The region between A199 and G206 expanded after 60 ns of simulation (Appendix A), which resulted in a more extended radius of gyration distribution. 

LaRP6-LaM has a long C-terminal IDR, containing 30 residues length located between T70-E90. Given this long IDR, AMBER99SB-ILDN simulations result in a collapsed conformation (average Rg = 1.44 nm), far from the experimental range. On the other hand, both simulations with CAIPi3P achieved radius of gyration distributions within the expected values (average Rg = 1.7 nm for AMBER99SB-ILDN+CAIPi3P and 1.69 nm for AMBER03ws+CAIPi3P). This also reflects on the χ^2^ values obtained for LaRP6-LaM simulations (Table 4), which show that only CAIPi3P simulations obtained values under the critical value for this protein (Figure 10, χ^2^ = 0.3 for AMBER99SB-ILDN+CAIPi3P and 1.1 for AMBER03ws+CAIPi3P). This difference comes from the sassed conformation of both C-Terminal and N-terminal loops, as shown in Appendix A. 

### 2.6. Applicability of CAIPi3P Solvation Model to Globular Proteins

To benchmark the CAIPi3P model, we simulated two model globular proteins; lysozyme and ubiquitin, using the same protocol as previously described for IDPs and partially unfolded At2g23090. For lysozyme and ubiquitin, the residual root mean square fluctuations (RMSF) obtained for both water models when applying established AMBER99SB-ILDN force field are shown in Appendix A, respectively. Simulations with both water models achieved very similar results, with only one region (loop 40–50) with markedly increased RMSF when applying the TIP3P model compared to CAIPi3P. Again, we attribute this difference to stronger electrostatic solvent–solute interactions in CAIPi3P, which increased the stability of the protein region, decreasing its overall per-residue RMSF. Yet the effect was much less pronounced for ubiquitin than for lysozyme.

## 3. Discussion

This work focused on the development of the novel three-point solvation model, denoted as CAIPi3P. Compared to the established and popular TIP3P model, CAIPi3P, which is based on the same framework, considerably improved the sampling of intrinsically disordered model peptides. All-atom MD simulations using CAIPi3P improved the SAXS scattering profile for two model IDPs: R/S peptide and histatin 5, and partially disordered At2g23090 from *A. thaliana* with the central IDR. The improvement was evident for all force fields used for the protein, although the selection of the most appropriate force field plays a vital role in the sampling improvement.

For the R/S peptide, the improvement was evident in simulations with the AMBER03ws force field. Application of the CAIPi3P model resulted in a better agreement for the radius of gyration since the framework prevented the artificial collapse of the polypeptide chain, which is a common pitfall of atomistic simulations of IDPs. CAIPI3P, due to modified electrostatics, maintained the generated stretched conformation, which resulted in better agreement with the experimental data. In the interpretation of the results, it is essential to focus on the differences in primary sequence between these two model IDPs. Histatin 5 has several polar residues dispersed throughout the length of the peptide, resulting in an overall uniform polar distribution. This homogeneous distribution helps the polypeptide chain to maintain favourable interactions with the solvent, resulting in the overall expanded structure. The R/S peptide is polar and charged, with the charged residues located within the eight C-terminal arginine–serine (R/S) repeats Figure 5, highlighted regions in the right panel). The obtained ensemble was affected by the C-terminal charge distribution, which facilitated the collapse of the polypeptide chain. Such a collapse was reduced when the AMBER03ws force field was applied. The sampling was further improved when CAIPi3P water was used, since it favoured the solute–solvent electrostatic interactions due to increased dipole moment of the water molecule. Solvent–solute interactions thus competed with excessive intramolecular solute–solute interactions, which led to the collapse.

In this work, only AMBER force fields were tested. Rauscher and co-workers [32] used R/S peptide to assess the accuracy of the CHARMM36m, obtaining accurate results for SAXS scattering profile.

For *At2g23090*, MD simulations showed a good agreement with experimental data when using the CAIPi3P water model in combination with the AMBER99SB-ILDN protein force field. Differences between AMBER99SB-ILDN and AMBER03ws lay within the side chain charge distribution and how certain residues interact with the solvents [28,39]. Consequently, in the *At2g23090* simulations, the compact globular C-terminal domain unfolded, increasing the interactions with the solvent molecules and the internal structural energy. In contrast, AMBER99SB-ILDN force field held the globular domains folded. This resulted in a similar SAXS pair distance distribution function (PDDF) between the resulting ensemble and the experimental data when using the CAIPi3P model. CAIPi3P water molecules interacted with the polar regions of the protein, improving the local sampling within the intrinsically disordered region and shielding the long-range interactions, avoiding the artificial collapse of the polypeptide chain. It is important to remark that *At2g23090* was solved by NMR using the ARIA [40] with explicit water refinement [41]. This method of refinement, albeit reliable, may bias the final ensemble towards a conformation towards states sampled with TIP3P. However, this structure was based not only on ARIA, but also NMR restraints, which should reduce bias on the final NMR ensemble and in our simulations. An evidence that the ARIA bias is not substantial is the fact that *At2g23090 TIP*3P simulations self-collapsed, resulting in an ensemble that differed significantly form the NMR conformations.

The average radius of gyration was also closer to the experimental value when CAIPi3P was used. Table 2 shows all the calculated and experimental values for all tested systems. Given the high structural fluctuations in IDPs, the error bars have a significant intersection. Hence, there is no statistical difference in this subject when TIP4P/2005 and CAIPi3P are compared for both histatin 5 and R/S peptide.

For the LaRP6-LaM and LaRP6-RRM1, we decided to focus on the most known combinations of force-field and water models used in this work, AMBER99SB-ILDN+TIP3P and AMBER03ws+TIP4P/2005, and how these force fields would be affected by CAIPi3P. Given the low extent of the disordered loops within LaRP6-RRM1, the usage of CAIPi3P did not cause a substantial improvement in comparison to TIP3P. However, simulations employing CAIPi3P sampled Rg values more accurately when combined with the AMBER03ws force field. LaRP6-LaM has a long IDR in its C-terminus, and because of this, CAIPi3P significantly improved the accuracy of the obtained conformations for both force fields, since it avoided the self-collapse of the C-terminal regions χ^2^ when used with AMBER99SB-ILDN and stabilized the structured scaffold of the core region when combined with AMBER03ws.

Nonetheless, there is a considerable improvement in the accuracy of the sampled conformations when simulations were carried out with the CAIPi3P solvation model. Table 4 shows that systems simulated with CAIPi3P resulted in the lowest difference between the calculated and experimental SAXS scattering profile, with the root mean square difference between the calculated and experimental PDDF shown in Table 4.

The bulk water parameters calculated for CAIPi3P are summarised in Table 5. By changing the dipole moment of the TIP3P water model most of the bulk water parameters were improved for CAIPi3P in comparison to the standard TIP3P model. However, several significant changes need to be addressed, such as the average oxygen-oxygen radial density distance (R_O-O_) and the density. The R_O-O_ distance for CAIPi3P was lower than the experimental distance, resulting in a higher density of 1.06 g/cm^3^. This results in a more compact water configuration, increasing the water-water correlation and decreasing the overall potential energy of the bulk water, and deeply affecting the density temperature dependence (Appendix A). Therefore, the usage of a higher dipole yields higher barriers to reorganise the solvent surrounding the solute, which contributes to the better sampling of the protein observed in CAIPi3P simulations.

The differences between experimental and CAIPI3P bulk water parameters shows that the latter require improvements. These modifications may come in tuning the vibrational frequency of the H-O-H angle to modify water-water interactions to decrease the magnitude of hydrogen bonds, which should yield better agreement with experimental data. CAIPi3P improved several different parameters when compared to the standard TIP3P model; however, it still less accurate in comparison to OPC and TIP4P/2005. Future work needs to address the bulk water issues of CAIPi3P model, which should further increase the applicability of CAIPi3P to different systems.

## 4. Materials and Methods 

To assess the role solvation effects have in reproducing the experimental parameters of IDPs, and to evaluate the applicability of the CAIPi3P model to studies of “mixed” ordered–disordered systems, we selected model IDPs (histatin 5 [30] and R/S peptide [32]) and At2g23090, which is partially disordered. To determine the performance of the model, simulations were made for a comparison between CAIPi3P and established water models. 

Fully extended conformations of histatin 5 (sequence: DSHAKRHHGYKRKFHEKHHSHRGY) and R/S-peptide (sequence: GAMGPSYGRSRSRSRSRSRSRSRS) were built using the UCSF Chimera [42] package since their experimental atomistic structures were not available. The conformational ensemble of *A. thaliana At2g23090* (PDB code: 1WVK), obtained by solution NMR, was used to calculate the small-angle X-ray scattering (SAXS) distribution and radius of gyration. The lowest-energy conformer was selected as a starting point for all-atom molecular dynamics (MD) simulations. 

For all systems investigated, missing hydrogen atoms were added, and several combinations of protein and water parametrisations were chosen, as summarised in Table 6. All simulations were performed using the Gromacs 5.3 suite [43]. The combinations of the force field and water models used are summarised in Table 7. For each combination, a 1 nm cubic box was centred on the structure. 

The system was solvated with the necessary number of water molecules to fill the protein simulation box. Next, sodium and chloride ions were added to the system at a concentration of 0.1 M to neutralise the simulation unit and to mimic the “physiological” salt concentration. The bonds were constrained using the LINCS algorithm [44], setting a 2 fs time step. The electrostatic interactions were calculated using the particle-mesh Ewald method [45], with a non-bonded cut-off set at 1 nm. All structures were energy minimised using the steepest descent algorithm for 20,000 steps. The minimisation was stopped when the maximum force fell below 1000 kJ/mol/nm using the Verlet cutoff scheme. This was followed by an equilibration run (NVT ensemble) of 20 ps with a time step of 2 fs and position restraints applied to the backbone, where the system was heated from 0 to 300 K; and another equilibration (NPT ensemble) at the constant temperature (300 K, 20 ps, 2 fs step) with backbone position restraints applied, and the constant pressure (1 bar). The temperature was set constant at 300 K by using an alternative Berendsen [46] thermostat (τ = 0.1 ps). The pressure was kept constant at 1 bar by using a Parrinelo–Rahman barostat with isotropic coupling (τ = 2.0 ps) to a pressure bath [47]. Finally, three production runs (NPT ensemble) of 100 ns were run for each system, using every force field–solvation model combination.

Ubiquitin (PDB code: 1UBQ) and lysozyme (PDB code: 253L) were selected for comparative runs to assess the effect of CAIPi3P water model on globular proteins with no IDRs. The simulation methodology was the same as the one described for the IDPs, with the exception that only the AMBER99SB-ILDN force field was used in combination with either the TIP3P or CAIPi3P solvation model. To check the convergence of simulation, the average radius of gyration and the radius of gyration standard deviation through time were plotted and analysed for histatin 5 (Appendix A), R/S Pep (Appendix A), *At2g23090* (Appendix A), LaRP6-RRM1 (Appendix A) and LaRP6-LaM (Appendix A). Since most of the simulations achieved a plateau within 100 ns, the sampled ensembles were used for the SAXS calculations.

CRYSOL [48,49] software was used to calculate the SAXS scattering patterns, along with the GNOM [48] software to calculate radial density distributions. The root square difference (RMSDexp-calc) between the experimental and the calculated were SAXS density made using an in-house script. The *gmx gyrate* module from the Gromacs suite was used to calculate the radii of gyration from the obtained trajectories. RMSF and RMSD values were calculated using the Gromacs suite (*gmx rms* and *gmx rmsf*, respectively). To evaluate the similarity between the distribution curves, their root square deviation (RMSD_exp-calc_) was calculated. To evaluate the accuracy for the sampled conformation in comparison to the experimental SAXS scattering, the reduced χ^2^ were calculated between the interpolated experimental dataset and the calculated scattering profiles for each simulation. The errors used to calculate χ^2^ values were based on the normalised average experimental error of 0.02.

The internal energy calculated in this work was made using GROMACS gmx energy, by calculating all bonded energy potentials (bonds, angles, dihedrals and improper dihedrals) a and non-bonded potentials (Coulombic potential and Lennard-Jones potential) for intramolecular interactions for each protein.

The bulk water properties were calculated using the protocols used by Izadi and co-workers [25].

## 5. Conclusions

To summarise, the parametrised dipole moment and partial atomic charges for the TIP3P water model generated a new solvation model denoted Charge-Augmented Three-Point Water Model for Intrinsically-Disordered Proteins (CAIPi3P). This model is transferrable, robust, and suitable for the atomistic MD simulations of IDPs, resulting in ensembles with a considerably better agreement with experimental data (SAXS). For the IDP models (histatin 5 and R/S peptide), simulations using CAIPi3P resulted in better agreement between calculated SAXS PDDFs and scattering profiles with experimental data than other simplified classical water models. CAIPi3P is also applicable to studies of globular proteins and—most importantly—functionally relevant multidomain proteins bearing globular domains and intrinsically disordered regions. The major shortcoming of CAIPI3P, which requires improvement, is its bulk water parameters. Although CAIPi3P model improved several different parameters when compared to the standard TIP3P framework, it still less accurate in comparison to OPC and TIP4P/2005. Future work needs to address it, which should further improve the model and increase its transferability.

## Figures and Tables

**Figure 1 ijms-21-06166-f001:**
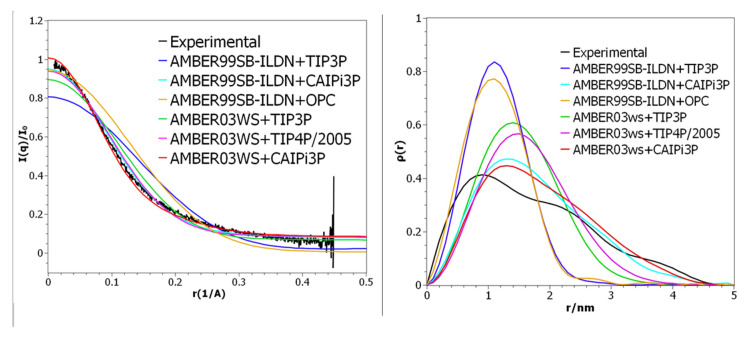
(**Left panel**) Small-angle X-ray scattering (SAXS) intensities, with a focus on the low angle region. (**Right panel**) Pair distance distribution function for histatin 5 SAXS.

**Figure 2 ijms-21-06166-f002:**
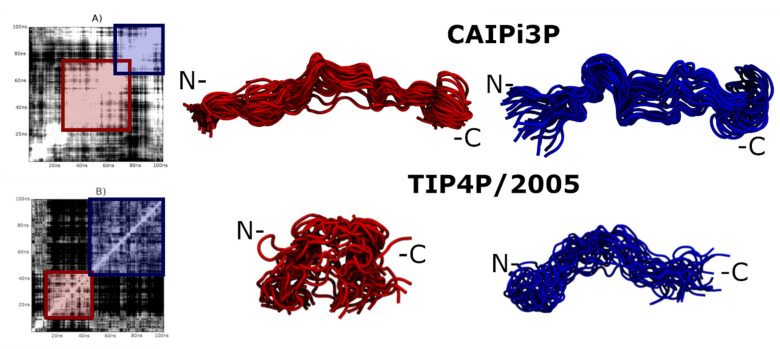
RMSD matrices and their respective clusters obtained by AMBER03ws. (**A**) CAIPi3P matrix. (**B**) TIP4P/2005 matrix. Red and blue molecular representations show the ensembles at the beginning and the end of the trajectory, respectively. The polypeptide chain did not collapse during the simulation using CAIPi3P water model (blue cluster in the top panel), in contrast to the clusters obtained from the simulations using TIP4P, which for the part of the trajectory remained collapsed (red cluster in the lower panel).

**Figure 3 ijms-21-06166-f003:**
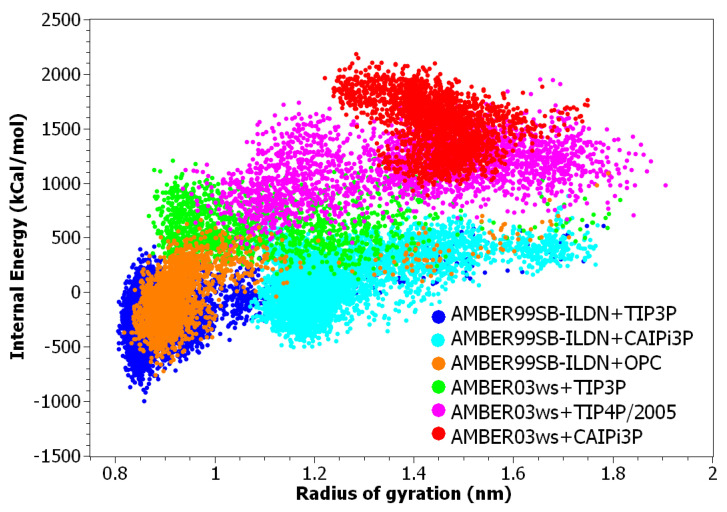
Histatin 5′s structural energy in the function of its radius of gyration. An increase in the structural energy is required to attain clusters in the experimental range of radius of gyration. Given the uniform distribution of charged residues throughout the sequence of histatin5, the increase in internal potential energy shows that CAIPi3P increases the solvent–solute interactions.

**Figure 4 ijms-21-06166-f004:**
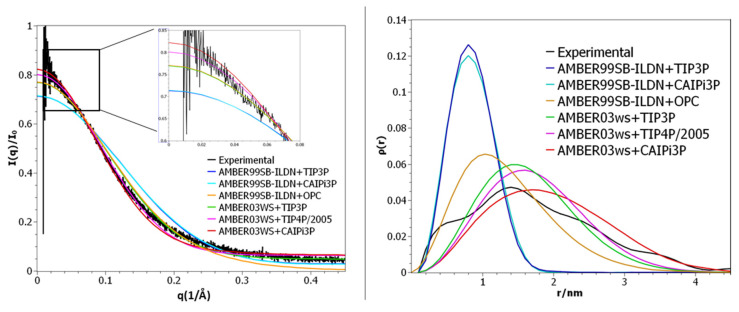
(**Left panel**) Small-angle X-ray scattering (SAXS) intensities. (**Right panel**) Pair distance distribution function for R/S peptide SAXS.

**Figure 5 ijms-21-06166-f005:**
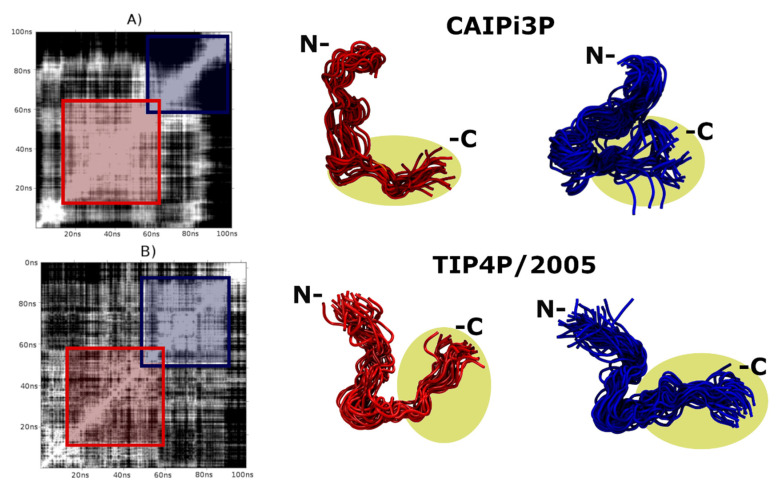
RMSD matrices for the R/S peptide and their respective clusters for AMBER03ws. (**A**) CAIPi3P matrix (**B**) TIP4P/2005 matrix. R/S repeat is circled and highlighted yellow.

**Figure 6 ijms-21-06166-f006:**
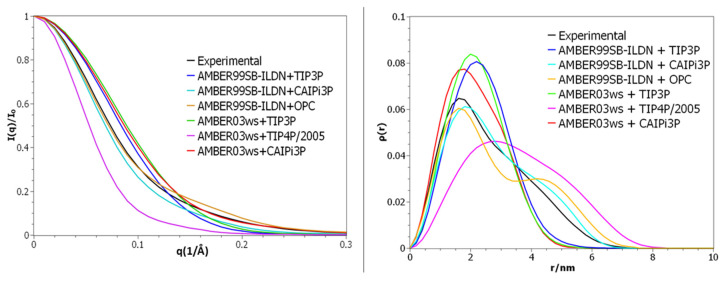
(**Left panel**) Small-angle X-ray scattering (SAXS) intensities. (**Right panel**) Pair distance distribution function for At2g23090 SAXS.

**Figure 7 ijms-21-06166-f007:**
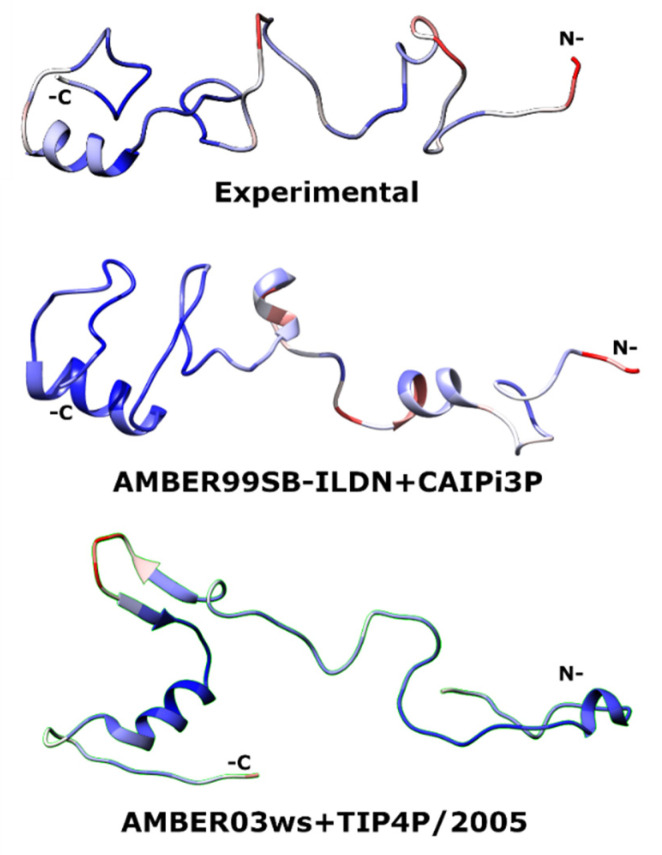
Average structures for the At2g23090. Highly flexible regions (high per-residue RMSF) are coloured red, while more rigid regions with lower per-residue RMSF are coloured blue.

**Figure 8 ijms-21-06166-f008:**
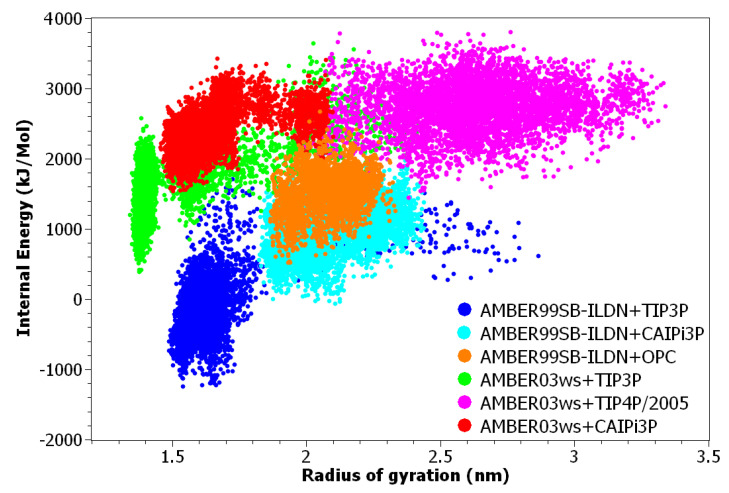
At2g23090 structural energy versus radius of gyration. The increase in structural energy gained by using AMBER03ws unfolded the structured region, stretching the average configuration. When using AMBER99SB-ILDN with CAIPi3P, the energy has been kept within the collapsed AMBER99SB-ILDN+TIP3P. The solvent model stabilised the unstructured sequences, and a structured biased force-field stabilised the intramolecular interactions sufficiently.

**Figure 9 ijms-21-06166-f009:**
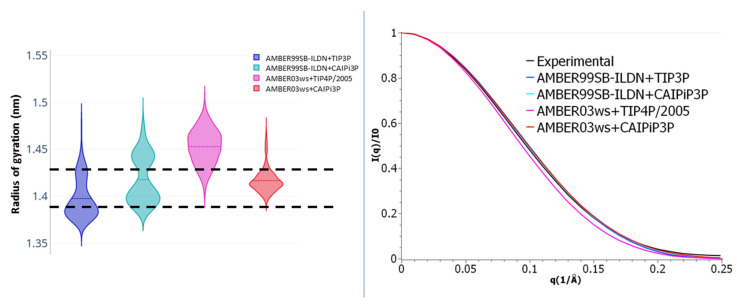
(**Left panel**) LaRP6-RRM1 radius of gyration violin plots; the experimental radius of gyration range is shown as dashed lines. (**Right panel**) Scattering profile for the LaRP6-RRM1 for simulations with different force field-water model combinations.

**Figure 10 ijms-21-06166-f010:**
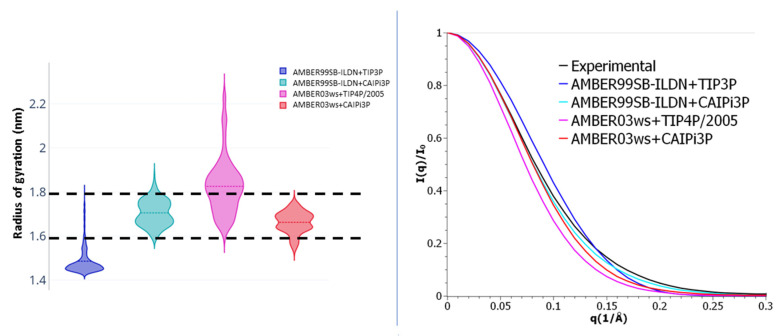
**Left panel**: LaRP6-LaM radius of gyration violin plots; the experimental radius of gyration range is shown by dashed lines. **Right panel**: scattering profile for the LaRP6-LaM for simulations with different force field–water model combinations.

**Table 1 ijms-21-06166-t001:** Partial atomic charges and resulting dipole moments for CAIPi3P, TIP3P, and TIP4P/2005 water models.

	O Charge (e)	Dummy Atom Charge (e)	H Charge (e)	Dipole Moment (D)
CAIPi3P	−0.954	-	0.477	2.69
TIP3P	−0.834 [25]	-	0.417 [25]	2.36 [25]
TIP4P/2005	-	−1.128 [29]	0.5564 [29]	2.30 [29]
Experimental	-		-	2.5−3 [25]

**Table 2 ijms-21-06166-t002:** Radius of gyration [in nm] for all macromolecules with all used force field and solvent combinations. The error values are the average standard deviation of the replicas. A99SB represents AMBER99SB-ILDN and A03ws represents AMBER03ws.

	A99SB+ TIP3P	A99SB+ CAIPi3P	A99S+OPC	A03ws+ TIP3P	A03ws+ TIP4P/2005	A03ws+ CAIPi3P	EXP
Histatin 5	0.7 ± 0.1	1.2 ± 0.2	0.9 ± 0.1	0.9 ± 0.2	1.4 ± 0.2	1.4 ± 0.1	1.3 ± 0.05
R/S-pep	0.9 ± 0.1	0.9 ± 0.2	1.1 ± 0.1	0.9 ± 0.2	1.2 ± 0.1	1.3 ± 0.2	1.3 ± 0.05
At2g23090	1.8 ± 0.2	2.1 ± 0.1	1.9 ± 0.2	1.7 ± 0.3	2.6 ± 0.2	1.9 ± 0.2	1.8 ± 0.2
LaRP6-RRM1	1.4 ± 0.05	1.4 ± 0.05	-	-	1.4 ± 0.05	1.4 ± 0.05	1.4 + 0.1
LaRP6-LaM	1.4 ± 0.1	1.7 ± 0.1	-	-	1.8 ± 0.2	1.6 ± 0.1	1.7 + 0.2

**Table 3 ijms-21-06166-t003:** Reduced χ^2^ and RMSD metric for I/I_0_ distribution for all tested molecules.

	AMBER99SB-ILDN+TIP3P		AMBER99SB-ILDN+CAIPi3P		AMBER99SB-ILDN+OPC		AMBER03ws+TIP3P		AMBER03ws+TIP4P/2005		AMBER03ws+CAIPi3P	
	χ^2^	RMSD	χ^2^	RMSD	χ^2^	RMSD	χ^2^	RMSD	χ^2^	RMSD	χ^2^	RMSD
Histatin 5	3.4	0.30	0.4	0.02	3.1	0.3	1.4	0.08	1.1	0.03	0.4	0.02
R/S-pep	5.6	0.10	5.7	0.14	3.5	0.07	2	0.05	1.45	0.03	1.3	0.03
At2g23090	3	0.06	1.0	0.1	0.4	0.01	5.3	0.10	20.6	0.42	3.7	0.08

**Table 4 ijms-21-06166-t004:** Reduced χ^2^ and RMSD metric for I/I_0_ distribution for LaRP6 LAM and LaRP6 RRM1.

	AMBER99SB-ILDN+TIP3P		AMBER99SB-ILDN+CAIPi3P		AMBER03ws+TIP4P/2005		AMBER03ws+CAIPi3P	
	χ^2^	RMSD	χ^2^	RMSD	χ^2^	RMSD	χ^2^	RMSD
LaRP6-LaM	1.6	0.02	0.3	0.01	4.4	0.03	1.1	0.01
LaRP6-RRM1	0.3	0.01	0.2	0.01	0.7	0.02	0.2	0.01

**Table 5 ijms-21-06166-t005:** Root-mean-square difference between experimental SAXS and calculated SAXS pair distance distribution.

	AMBER99SB-ILDN+TIP3P	AMBER99SB-ILDN+CAIPi3P	AMBER03ws+TIP3P	AMBER03ws+TIP4P/2005	AMBER03ws+CAIPi3P
Histatin 5	0.01	0.01	0.01	0.01	0.006
R/S-pep	0.03	0.03	0.01	0.008	0.007
At2g23090	0.01	0.005	0.01	0.014	0.014

**Table 6 ijms-21-06166-t006:** Bulk water parameters calculated for CAIPi3P. These were calculated using the methods explained in Izadi and coworkers [25].

	CAIPi3P	TIP3P	TIP4P/2005	OPC	Experimental
Dipole moment (μ (D))	2.69	2.34	2.305	2.48	2.5–3
Density (g/cm^3^)	1.05 ± 0.05	0.980	0.993	0.997	0.997
∆H_vap_[kcal/mol]	10.6 ± 0.05	10.26	10.89	10.57	10.52
Isobaric Heat Capacity *C*_p_ [cal/(K mol)]	23.7 ± 0.5	18.7	18.9	18	18
Thermal expansion α[10^−4*K^−1]	5.4 ± 0.1	9.2	2.8	2.7	2.56
O-O first peak distance [Å]	2.7	2.77	2.78	2.8	2.8
Static dielectric constant [ϵ_0_]	74.5 ± 1	94	60	78	78.4
Self-diffusion coefficient [m^2^/s]	4.67±0.2	5.5	2.08	2.3	2.3
Shear viscosity [cP]	1.1±0.1	0.321	0.855	-	0.896

**Table 7 ijms-21-06166-t007:** Macromolecules simulated and their respective force field/solvent combinations.

Protein	PDB Code	Force Field	Water Model
Histatin 5	-	AMBER99SB-ILDN [39]	TIP3P; CAIPi3P; OPC [25]
		AMBER03ws [27]	TIP3P; CAIPi3P; TIP4P/2005 [28]
R/S peptide	-	AMBER99SB-ILDN	TIP3P; CAIPi3P; OPC
		AMBER03ws	TIP3P; CAIPi3P; TIP4P/2005
*At2g23090*	1WVK	AMBER99SB-ILDN	TIP3P; CAIPi3P; OPC
		AMBER03ws	TIP3P; CAIPi3P; TIP4P/2005
LaRP6-RRM1	2MTG	AMBER99SB-ILDN	TIP3P; CAIPi3P
		AMBER03ws	CAIPi3P; TIP4P/2005
LaRP6-RRM1	2MTF	AMBER99SB-ILDN	TIP3P; CAIPi3P
		AMBER03ws	CAIPi3P; TIP4P/2005

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
