# Peer review of "Development of Charge-Augmented Three-Point Water Model (CAIPi3P) for Accurate Simulations of Intrinsically Disordered Proteins"

_ijms, 2020, doi:10.3390/ijms21176166_

Round 1
Reviewer 1 Report
The authors tried to improve a three-point water model for intrinsically disordered proteins. The manuscript might be publishable, but currently there are problems with both the claims and presentations. Briefly the current data does not justify that charge-augment is the key to improve the solvation and therefore conformational ensemble of disordered proteins. The authors might want to provide additional evidence with better error analysis and improve their presentation.
One major problem of the manuscript is the claim that CAIPi3P increases the solvent-solute interactions through tuning partial charges. However the authors still used the protein-water scaling LJ parameters to pair with CAIPi3P, then it is not clear if partial charge or LJ scaling is the key ingredient in capturing the solvation in the simulations. The question then would be, if without scaling the water-protein interaction, can CAIPi3P model still work for disordered proteins? Such a control simulation would tell if partial charge is really important in correcting the dimension of IDPs in all-atom simulations or it is just a secondary term.
The current comparison between simulation and experiment can be problematic. First, it does not take into account both the errors of experiments and simulations. Chi square instead of RMSD should be used, since these force fields might all be equally good or bad and is not distinguishable due to errors. This is likely the case considering the large errors of Rg (~20%) in simulations. Second, determining PDDR from SAXS data is tricky considering inverse Fourier transformation. A more straightforward way is to calculate SAXS intensity using all-atom data and compare with experimental I(q). Experimental errors of I(q) can also be directly used for chi square calculation.
Another major problem of building a water model based on TIP3P is the viscosity of the water even though computationally it is more efficient than TIP4P. What is the viscosity of the new CAIPi3P model using the shear viscosity method (Gonzalez and Abascal J Chem. Phys. 2010, 132:096101)? The authors might also want to provide the bulk water parameters of TIP4P/2005 in Table 3. Most of those values can be found in previous literature. The other aspect of the water model is the temperature dependent density, TIP4P/2005 is known to reproduce the experiment density well. What about CAIPi3P?
The caption of Fig. 1 is confusing. It seems Fig. 1 is showing the PDDR (pair distance distribution function) of Histatin 5 in different force fields. CAIPi3P + 03ws has a more broad distribution in pddr than TIP4P/2005 + 03ws. However in Fig. 3, Rg from CAIPi3P seems to cover a smaller range of values.
How is the structural energy calculated from all-atom simulation? The electrostatic and van der Waals energies from all-atom simulation do not take into account the entropic contribution. Calculating the solvation free energy of side chain analogues of 20 amino acids and comparing with experimental data might tell if CAIPi3P has larger solute-solvent interactions. Actually by looking at the Rg from different force fields, TIP4P/2005 and CAIPI3P are within error bars. So it is difficult to imagine there will be a big difference in solvation between TIP4P/2005 and CAIPI3P. By the way, the magnitudes of some of the errors of Histatin 5 in Table 2 are not correct. Please also check if the Rg (1.3nm) of At2g23090 from CAIPi3P+03ws in Table 2 is correct.
The authors might want to provide the details how they obtain the new partial charges for CAIPi3P, for example, the plot of objective function as a function of the parameters during the optimization.
Validation of convergence (error analysis or time correlation of Rg) should be provided. 100 ns of productive simulations are very short even for these small peptides.
A comma instead of a period is used in many of the values of RMSD to represent decimal point.
Author Response
On behalf of my coworkers and myself, I would like to thank all the reviewers for assessing our manuscript (ID: ijms-838798) and for providing some very helpful comments. All the corrections suggested by both reviewers were applied through the revised manuscript. Responses and clarifications to further specific comments are given below.
Reviewer 1
“The authors tried to improve a three-point water model for intrinsically disordered proteins. The manuscript might be publishable, but currently there are problems with both the claims and presentations. Briefly the current data does not justify that charge-augment is the key to improve the solvation and therefore conformational ensemble of disordered proteins. The authors might want to provide additional evidence with better error analysis and improve their presentation.
One major problem of the manuscript is the claim that CAIPi3P increases the solvent-solute interactions through tuning partial charges. However the authors still used the protein-water scaling LJ parameters to pair with CAIPi3P, then it is not clear if partial charge or LJ scaling is the key ingredient in capturing the solvation in the simulations. The question then would be, if without scaling the water-protein interaction, can CAIPi3P model still work for disordered proteins? Such a control simulation would tell if partial charge is really important in correcting the dimension of IDPs in all-atom simulations or it is just a secondary term.”
In the development of CAIPi3P model we have changed the dipole moment and therefore charges, not the Lennard-Jones term. The L-J modifications within CAIPi3P arise form TIP4P/2005 and AMBER03ws models. This has been clarified in the revised manuscript.
“The current comparison between simulation and experiment can be problematic. First, it does not take into account both the errors of experiments and simulations. Chi square instead of RMSD should be used, since these force fields might all be equally good or bad and is not distinguishable due to errors. This is likely the case considering the large errors of Rg (~20%) in simulations. Second, determining PDDR from SAXS data is tricky considering inverse Fourier transformation. A more straightforward way is to calculate SAXS intensity using all-atom data and compare with experimental I(q). Experimental errors of I(q) can also be directly used for chi square calculation.”
We have integrated I(q)/I0 plots in the revised manuscript. Also, their chi2 plots were generated from all simulations.
“Another major problem of building a water model based on TIP3P is the viscosity of the water even though computationally it is more efficient than TIP4P. What is the viscosity of the new CAIPi3P model using the shear viscosity method (Gonzalez and Abascal J Chem. Phys. 2010, 132:096101)? The authors might also want to provide the bulk water parameters of TIP4P/2005 in Table 3. Most of those values can be found in previous literature. The other aspect of the water model is the temperature dependent density, TIP4P/2005 is known to reproduce the experiment density well. What about CAIPi3P?”
The shear viscosity and density temperature dependency were added to the revised manuscript (Table 5 and Supp. Fig. 10, respectively).
“The caption of Fig. 1 is confusing. It seems Fig. 1 is showing the PDDR (pair distance distribution function) of Histatin 5 in different force fields. CAIPi3P + 03ws has a more broad distribution in pddr than TIP4P/2005 + 03ws. However in Fig. 3, Rg from CAIPi3P seems to cover a smaller range of values.”
These have been clarified in the revised version of the manuscript.
“How is the structural energy calculated from all-atom simulation? The electrostatic and van der Waals energies from all-atom simulation do not take into account the entropic contribution. Calculating the solvation free energy of side chain analogues of 20 amino acids and comparing with experimental data might tell if CAIPi3P has larger solute-solvent interactions. Actually by looking at the Rg from different force fields, TIP4P/2005 and CAIPI3P are within error bars. So it is difficult to imagine there will be a big difference in solvation between TIP4P/2005 and CAIPI3P. By the way, the magnitudes of some of the errors of Histatin 5 in Table 2 are not correct. Please also check if the Rg (1.3nm) of At2g23090 from CAIPi3P+03ws in Table 2 is correct.”
The Rgs have been amended.
“The authors might want to provide the details how they obtain the new partial charges for CAIPi3P, for example, the plot of objective function as a function of the parameters during the optimization.”
The optimisation graph is now included in the supporting data.
“Validation of convergence (error analysis or time correlation of Rg) should be provided. 100 ns of productive simulations are very short even for these small peptides.”
The convergence of the average radius of gyration and the convergence of the radius of gyration standard deviation were added to the supporting data. Most of the simulations achieved the plateau within the 100 ns time scale.
“A comma instead of a period is used in many of the values of RMSD to represent decimal point.”
This has been resolved in the revised manuscript.
Reviewer 2 Report
The manuscript by de Souza et al., entitled with "Development of Charge-Augmented Three-Point Water Model (CAIPi3P) for Accurate Simulations of Intrinsically Disordered Proteins" is a challenging and potentially important study about the improvement / development of water model for molecular dynamic simulation. The authors seem to succeed in developing a new water model called CAIPi3P, with relevant improvement of MD simulation of intrinsically disordered proteins. The subject matter of this work is laudable and of interest to both the protein MD community and intrinsically disordered protein community. However, the reviewer thinks that experimental design of this work still have some non-negligible problems as indicated below.
(1) Except at2g23090, the authors selected two samples (histatin 5 and R/S-peptide) without any bioinformatic prediction whether the sequences may contain IDR or not. This situation is difficult to imagine since the water model is designed for IDP calculation as the authors mentioned. This reviewer is very interested how these proteins are appeared in their IDP propensity.
(2) For the simulation and its interpretation of the result of at2g23090, the selection of the entry PDB 1WVK. Although the original article for this entry and the solution structure of at2g23090 has not yet been published yet, in the PDB entry, the method of NMR refinement of this entry was described as "refined by ARIA with explicit solvent". This means that the structural ensemble of this PDB entry was determined not only by NMR experimental constraints, such as NOEs and dihedral angles, but also with some energetic calculation during simulation.
The explicit solvent refinement method was developed by Dr. Jens P Linge and co-authors (doi : 10.1002/prot.10299), and the refinement method is largely biased with the force field PARALLHDG and water model TIP3P.
Thus the authors may try first with the other NMR structure entry without this highly biased refinement method but without ARIA refinement.
Then re-evaluate this simulation result with why TIP3P-originated NMR structure ensemble fit better for CAIPi3P model rather than TIP3P control parameter set. Otherwise, the authors' claim that "CAIPi3P solvation model is a valuable tool for molecular simulations of intrinsically disordered proteins" is hardly trusted.
Author Response
On behalf of my coworkers and myself, I would like to thank the reviewers for assessing our manuscript (ID: ijms-838798) and for providing some very helpful comments. All the corrections suggested by both reviewers were applied through the revised manuscript. Responses and clarifications to further specific comments are given below.
Reviewer 2
“The manuscript by de Souza et al., entitled with "Development of Charge-Augmented Three-Point Water Model (CAIPi3P) for Accurate Simulations of Intrinsically Disordered Proteins" is a challenging and potentially important study about the improvement / development of water model for molecular dynamic simulation. The authors seem to succeed in developing a new water model called CAIPi3P, with relevant improvement of MD simulation of intrinsically disordered proteins. The subject matter of this work is laudable and of interest to both the protein MD community and intrinsically disordered protein community. However, the reviewer thinks that experimental design of this work still have some non-negligible problems as indicated below. (1) Except at2g23090, the authors selected two samples (histatin 5 and R/S-peptide) without any bioinformatic prediction whether the sequences may contain IDR or not. This situation is difficult to imagine since the water model is designed for IDP calculation as the authors mentioned. This reviewer is very interested how these proteins are appeared in their IDP propensity.”
The bioinformatics prediction would not be required, given both systems investigated in our manuscript (histatin 5 and R/S peptide) have been studied extensively for their IDP characteristics and are CONFIRMED intrinsically disordered peptides: references 28,29 and 30 in our manuscript clearly show this.
“For the simulation and its interpretation of the result of at2g23090, the selection of the entry PDB 1WVK. Although the original article for this entry and the solution structure of at2g23090 has not yet been published yet, in the PDB entry, the method of NMR refinement of this entry was described as "refined by ARIA with explicit solvent". This means that the structural ensemble of this PDB entry was determined not only by NMR experimental constraints, such as NOEs and dihedral angles, but also with some energetic calculation during simulation.”
This is a very valid point and we are very grateful to the reviewer for highlighting it. A clarification has been added to the revised manuscript.
“The explicit solvent refinement method was developed by Dr. Jens P Linge and co-authors (doi : 10.1002/prot.10299), and the refinement method is largely biased with the force field PARALLHDG and water model TIP3P. Thus, the authors may try first with the other NMR structure entry without this highly biased refinement method but without ARIA refinement.
Then re-evaluate this simulation result with why TIP3P-originated NMR structure ensemble fit better for CAIPi3P model rather than TIP3P control parameter set. Otherwise, the authors' claim that "CAIPi3P solvation model is a valuable tool for molecular simulations of intrinsically disordered proteins" is hardly trusted.”
This is yet another important point – to address it, we have added the following section (p. 12): “It is important to note that the NMR structure of At2g23090 was solved using the ARIA method[38]. This method of refinement, albeit reliable, may bias the final ensemble towards a conformation towards states sampled with TIP3P. However, given the fact that the solution of this structure was based not only on ARIA but also NMR restraints which should reduce bias on the final NMR ensemble and in our simulations. An evidence that the ARIA bias is not substantial is the fact that At2g23090 TIP3P simulations produced a self-collapsed ensemble, which differs significantly form the NMR conformations.”
In addition, to thoroughly address the reviewer’s comments, we have added another partially disordered protein to the set: human RNA-binding protein LaRP6. Its NMR structure (PDB codes: 2MTF and 2MTG) have not employed ARIA, and the results, i.e. an improvement of sampling upon CAIPi3P application has been observed, consistently with the results obtained for At2g23090.
We hope that our response thoroughly address all the concerns regarding the results, data analysis and presenting of the data, and we are looking forward to the feedback.
Round 2
Reviewer 1 Report
I acknowledge the authors' efforts in preparing this revision and adding additional data. However problems still exist. I would not recommend the manuscript to publish in IJMS and suggest the authors to reconsider their motivation of making this water model and the difference with existing water models.
Here are my detailed comments:
1. Regarding the question of partial charge and LJ parameters, I understand the authors changed the partial charge but LJ terms. However water scaling is in Amber03ws. My question was if no water scaling was introduced, do they still obtain the same improvement for simulating IDP. This is important since if their partial charge modification cannot improve the simulation without water scaling, then it is difficult to convince people why these partial charge modification is needed for simulating IDPs. I have to note that these partial charge modification do not make water model itself a better water model without the context of simulating IDPs. Then the authors have to seriously consider their motivation of making such a water model. In fact, there have been several other work of improving the water model without the need of scaling protein-water interactions, for example TIP4PD (Piana & Shaw) and the OPC water model (Onufriev). The OPC water in my opinion did what the authors would like to in improving simulation of IDPs and more in improving the bulk water properties. A comparison with the OPC model is necessary.
2. The values of chi square provided are confusing. I guess the authors did the summation of squared difference divided by the squared errors instead of the mean. Since the authors mentioned there are 51 data points, I would divide them by 51 and consider these chi square values. Since all these chi square values are close to 1 when comparing with SAXS, one cannot really tell which force field is better than the others considering the experimental and simulation errors. I am also not sure if simulation errors are included in the chi square calculation since there seems to be significant errors in Rg (Fig. S3-12). 100 ns is clearly too short for these simulations after seeing these supporting figures.
3. It is nice to see that the shear viscosity is improved quite a lot for a 3 point water model, however the density of the new model is off quite a lot, raising serious doubts of the potential application of the water model.
Minor problem, I see many 'Error! References source not found'.
Author Response
Responses and clarifications to further specific comments by Reviewer 1 are given below:
- Regarding the question of partial charge and LJ parameters, I understand the authors changed the partial charge but LJ terms. However water scaling is in Amber03ws. My question was if no water scaling was introduced, do they still obtain the same improvement for simulating IDP. This is important since if their partial charge modification cannot improve the simulation without water scaling, then it is difficult to convince people why these partial charge modification is needed for simulating IDPs. I have to note that these partial charge modification do not make water model itself a better water model without the context of simulating IDPs. Then the authors have to seriously consider their motivation of making such a water model. In fact, there have been several other work of improving the water model without the need of scaling protein-water interactions, for example TIP4PD (Piana & Shaw) and the OPC water model (Onufriev). The OPC water in my opinion did what the authors would like to in improving simulation of IDPs and more in improving the bulk water properties. A comparison with the OPC model is necessary.
As we have stated in the introduction, our reason for choosing simple, non-polarisable model in the development of CAIPi3P solvation model was its performance and popularity of three-point framework within the community (lines 62-68). As suggested, we have performed additional simulations using OPC model. As showed in figures and tables in the main manuscript and in supplementary data, the performance varied and was dependent on the system: both OPC and CAiPI3P models performed well in simulations of partially disordered protein, CAIPI3P outperformed all models, including OPC, in simulations of histatin 5. These results and their discussion has been included in the revised manuscript. Also, the references for OPC models (40 and 41) have been added to the list.
- The values of chi square provided are confusing. I guess the authors did the summation of squared difference divided by the squared errors instead of the mean. Since the authors mentioned there are 51 data points, I would divide them by 51 and consider these chi square values. Since all these chi square values are close to 1 when comparing with SAXS, one cannot really tell which force field is better than the others considering the experimental and simulation errors. I am also not sure if simulation errors are included in the chi square calculation since there seems to be significant errors in Rg (Fig. S3-12).
The errors we used to calculate χ2 values were based on the normalised average experimental error of 0.02. Given the fact that all force fields have similar range of standard deviation regarding the radius of gyration (Rg of ~0.2 nm), we decided to avoid using the associated error values for the simulations given the highly flexible nature of IDPs, focusing on the average values compared to the experimental associated error.
- It is nice to see that the shear viscosity is improved quite a lot for a 3 point water model, however the density of the new model is off quite a lot, raising serious doubts of the potential application of the water model.
We have commented on the shear viscosity and other parameters presented in Table 5 in the discussion (highlighted text, lines 395-401) and in the conclusions (highlighted text, lines 475-479). While we consider the shear viscosity of 1.1 ± 0.1 as a very significant improvement compared to 0.321 for TIP3P and 0.896 experimental value, some bulk parameters of CAIPi3P model clearly need further improvement, as stated in the discussion and in the conclusions. Suggested modifications may come in tuning the vibrational frequency of the H-O-H angle to modify water-water interactions to decrease the magnitude of hydrogen bonds, which should yield better agreement with experimental data. Nevertheless, considering its simplistic framework, which is rooted in one of the most popular models (TIP3P) among the protein simulations community, we consider CAIPI3P a good trade-off between the performance and the accuracy.
Minor problem, I see many 'Error! References source not found'.
This has been fixed.
We hope that our response thoroughly address all the concerns regarding the results, data analysis and presenting of the data, and we are looking forward to the feedback.
Reviewer 2 Report
After revision process, the manuscript has been much improved with the additional two simulations of the RRM domain case.
Only one point I like to recommend making correction as follows,
page 13, line 367-369.
It is important to note that the NMR structure of At2g23090 was solved using the ARIA method[38].
could be
... using ARIA method [ref] with "explicit solvent refinement option" [38].
Ref : J Mol Biol, 1997 Jun 13;269(3):408-22.
doi: 10.1006/jmbi.1997.1044. PMID: 9199409.
ARIA is the name of the software suits of NMR structural determination with broad functions. The problem may occur only when the users intend to apply "explicit solvent refinement option".
Author Response
On behalf of my coworkers and myself, I would like to thank you for assessing the revised version our manuscript (ID: ijms-838798). All the corrections suggested were applied through the revised manuscript. As you have suggested, the sentence about ARIA refinement (with the reference) has been restructured (lines 339-340 of the current manuscript).
Round 3
Reviewer 1 Report
Though some of the questions remain, I appreciate the authors' efforts and will leave them here.
The contribution between partial charge and LJ terms is still not addressed. The authors added the comparison with the OPC model. One motivation I agree with is that their model still follows the traditional geometry constraints of water. Then the advantage of their model besides better reproducing dielectric properties might be some of the water interactions requiring such geometry restraints (e.g. formation of hydrogen bond, water hydration shell ...). The authors might also want to check Carlos Vega's papers on the topic of scaled charge force field, which pointed out the possible inaccuracy of using classical force field in simulating dielectric constant. Maybe it's not simply about tuning the partial charge, instead one has to also calculate these properties using a different way due to the lack of polarizability in classical force field.
Regarding the chi2 calculation, the error of SAXS usually increases with increasing q range. So a normalized error might be problematic depending on the measurement. It is also a known problem that the SAXS intensity above qRg>3 might not be well reproduced when calculating the SAXS intensity from only protein coordinates. The highly flexible nature of IDP does not justify the fact that simulation error is not considered in chi2 calculation, since these errors are dependent on the convergence of the simulation. However it might be true that all simulations are with similar length and therefore similar errors, so they all contribute the same way to the chi2 values.
The authors' intuition on the way of further optimizing the bulk properties is in line with releasing the geometry restraints.